# Development of a novel EV-A71 monoclonal antibody for monitoring vaccine potency

Thi-Hong-Loc Le[1,2], Tzu-Yu Weng[1], Hua Yen[1], Min-Yuan Chia[1,3], Min-Shi Lee ᴵᴰ[1]*

1 National Institute of Infectious Diseases and Vaccinology, National Health Research Institutes, Zhunan, Miaoli, Taiwan, 2 Institute of Bioinformatics and Structural Biology, College of Life Sciences and Medicine, National Tsing Hua University, Hsinchu, Taiwan, 3 Department of Veterinary Medicine, College of Veterinary Medicine, National Chung Hsing University, Taichung, Taiwan

* minshi@nhri.edu.tw

## Abstract

Enterovirus A71 (EV-A71) is one of the major causes of hand, foot, and mouth disease (HFMD), primarily affecting children under five. It can lead to neurological and cardiac complications, or even death, in some cases. Inactivated monovalent vaccines have been licensed in China and Taiwan; however, the stability of EV-A71 vaccines is often compromised by factors such as extreme temperatures or ultraviolet (UV) irradiation. Currently, no commercially available tools can assess the stability of EV-A71 throughout vaccine development. In this study, we report the development of a monoclonal antibody (mAb), NHRI2016–1, which can be used in in vitro immunoassays to evaluate EV-A71 vaccine potency and effectiveness. NHRI2016–1 exclusively recognizes effective EV-A71 antigens in in vitro potency assays. Similarly, rat experiment confirmed that effective vaccine antigens could induce neutralizing antibodies, while ineffective antigens could not. Thus, NHRI2016–1 shows potential for correlating in vitro potency with in vivo immunogenicity of EV-A71 vaccine antigens. These data suggest that NHRI2016–1 could be a promising tool for characterizing EV-A71 vaccines and monitoring vaccine potency.

## Author summary

EV-A71 is the causative agent of hand, foot, and mouth disease (HFMD), which can occasionally progress to severe neurological complications and even death in young children. Inactivated monovalent EV-A71 vaccines have recently been marketed in China and Taiwan. However, vaccine potency can be permanently and irreversibly lost when exposed to extreme heat or UV light. Currently, no commercial tools are available to assess the stability of the EV-A71 vaccine. Therefore, developing an antibody to monitor vaccine potency is essential — not only to reduce the use of experimental animals, but also to minimize the wastage of both licensed and under-development vaccines due to potency loss.

**Data availability statement:** All relevant data are within the manuscript and its Supporting Information files.

**Funding:** This work was supported by the National Health Research Institutes (grant number 13A1-IVPP09 to MSL) and the National Science and Technology Council of Taiwan (grant number NSTC 112-2823-8-4000-001 to MSL). The funders had no role in study design, data collection and analysis, decision to publish, or preparation of the manuscript.

**Competing interests:** The authors have declared that no competing interests exist.

Here, we developed the monoclonal antibody NHRI2016–1, which can be used to quantitatively monitor vaccine potency in vitro and reflects its correlation with the in vivo immunogenicity of the EV-A71 vaccine antigen. This antibody-based approach offers a rapid, reproducible, and ethical alternative to traditional animal-based potency tests. Furthermore, the development of NHRI2016–1 represents a critical step toward ensuring consistent vaccine quality and safeguarding public health, particularly in regions where HFMD outbreaks are common.

## Introduction

EV-A71 is the predominant pathogen that primarily causes hand, foot, and mouth disease (HFMD). EV-A71 often co-circulates with other HFMD-causing agents such as coxsackievirus A6 (CVA6), coxsackievirus A10 (CVA10), and coxsackievirus A16 (CVA16). Among all human enteroviruses, EV-A71-related HFMD is chiefly identified in severe complications and fatal cases, accounting for 70% and 90%, respectively [1]. In addition, young children under the age of five are particularly susceptible to HFMD [2,3].

The EV-A71 genome is a single-stranded RNA encapsulated in a non-enveloped capsid virion with a length of roughly 7400 bases. The single open reading frame encodes a large polyprotein that is subsequently cleaved into structural capsid protein P1 and non-structural proteins P2 and P3. Capsid protein P1 is processed to produce four viral capsid proteins (VPs): VP1, VP2, VP3, and VP4. P2 and P3 are further processed into non-structural proteins 2A–2C and 3A–3D, respectively. Sixty copies of four VPs assemble into the EV-A71 capsid, in which VP1, VP2, and VP3 together are exposed on the viral capsid surface, while VP4 is entirely present inside the capsid [4].

EV-A71 is genetically divided into seven genogroups, A–G, based on phylogenetic analysis of the VP1 gene. The B and C genogroups are further classified into ten genotypes (B1–B5, C1–C5) [5,6]. Several HFMD outbreaks caused by B4, B5, and C4 genotypes have been reported in the Asia-Pacific region, including China, Japan, Malaysia, Singapore, Taiwan, and Vietnam, with a significant increase in morbidity rates in the past twenty-five years [7–12]. Therefore, developing multivalent vaccines demonstrating cross-protection against multiple EV-A71 genotypes has become increasingly important in preventing HFMD outbreaks.

Inactivated monovalent EV-A71 vaccines targeting the B4 and C4 genotypes have already been licensed and commercially used in Taiwan and China, respectively. These B4 and C4 genotype-based vaccines have demonstrated both in vitro and in vivo cross-reactivity against other genotypes [13,14]. As reported by Hong et al., from 2017 to 2019, following the launch of the EV-A71 vaccine in China, the proportion of HFMD patients infected with EV-A71 slightly decreased to 24.7%, while those infected with CVA16 and other enteroviruses increased slightly to 23.7% and 51.6%, respectively [15]. Additionally, Qiao et al. found that the predominant circulating enteroviruses serotypes have evolved over time. From 2007 to 2012, EV-A71 was

the dominant serotype in Asia; however, after 2013, CVA6 surpassed EV-A71 as the leading serotype in terms of prevalence. In Europe, EV-A71 and CVA6 exhibited dynamic epidemic trends from 2009 to 2018, while in Africa, CVA16 and EV-A71 were the main epidemic serotypes from 2007 to 2011. In the Americas, EV-A71 remained highly prevalent from 2005 to 2019, with coxsackievirus strains showing more intermittent patterns [16]. Despite these shifts, the ongoing circulation of EV-A71 and its potential to cause severe disease underscores the need for continued monitoring and vaccine development.

Naturally, EV-A71 cultured for vaccine development can produce two types of distinct antigenic states: highly infectious native virions (F particles) and expanded virions (E particles). The conversion of native virions into expanded virions is triggered under certain conditions, such as low pH and receptor engagement [17]. These antigenic forms are immunogenic [18]. However, exposure to extreme cold, heat, UV light, or insufficient excipients can cause a loss of vaccine potency that cannot be restored, and in some cases, even under optimal conditions, vaccine potency may still degrade gradually over time [19]. In this study, we assume vaccine antigens that retain their integrity and potency are effective antigens, while ineffective antigens refer to those that lose their potency. There is one commercial antibody, namely MAB979, which has been widely used for the detection and quantification of EV-A71 [20]. However, MAB979 binds to the linear epitope in the VP2 capsid protein and cannot differentiate between effective and ineffective antigens [20]. Therefore, developing antibodies that exclusively recognize effective antigens is desirable and critical for monitoring vaccine potency during development, manufacturing and product release.

Here, we developed a murine mAb, NHRI2016–1, which can be employed in antibody-based immunoassays, particularly dot blot and ELISA, to quantitatively distinguish between effective and ineffective EV-A71 antigens. NHRI2016–1 could serve as a valuable tool for assessing the correlation between in vitro potency and in vivo immunogenicity. Additionally, we identified VP3-S64R as a key amino acid involved in the binding site of NHRI2016–1, which in turn may have significant implications for the development of EV-A71 vaccines.

## Materials and methods

### Ethics statement

All animal experiments in this study were conducted in accordance with the guidelines of the Laboratory Animal Center of the National Health Research Institutes (NHRI, Taiwan). The rat immunization protocol was reviewed and approved by the NHRI Institutional Animal Care and Use Committee (approved no. NHRI-IACUC-113120-A). For the mouse monoclonal antibody generation, this study was contracted to a Contract Research Organization (CRO) - LTK Biolaboratories and approved (approved no. NHRI-IACUC-1514 and LTK-IACUC-234). After the completion of the experimental protocol, all the tested animals were euthanized by 100% $CO_2$ inhalation for 5 min, followed by cervical dislocation to minimize the suffering.

### Viruses and cells

An EV-A71 B5 genotype, namely B5-141 (GenBank accession number JN874552), was isolated from a herpangina patient at twenty months of age in 2008 [21]. High-growth B5-141-6-5 (HG-B5-141-6-5) virus was selected after a total of 28 passages and 2 times plaque selection using B5-141 [18]. The full-length of HG-B5-141-6-5 cDNA was cloned into the pcDNA3.1(-) vector and electroporated into Vero cells for virus generation. The rescued virus was named rgHGB5.

African green monkey kidney (Vero cells, ATCC CCL-81) cells and human Caucasian embryo rhabdomyosarcoma (RD cells, ATCC CCL-136) cells were used to propagate viruses. RD cells were maintained in Dulbecco's Minimum Essential Medium (DMEM, Gibco) supplemented with 10% v/v fetal bovine serum (FBS, Gibco) at 37°C±0.3°C in 5% $CO_2$ incubators (Thermo Fisher Scientific). Vero cells were cultured at 37°C±0.3°C in 5% $CO_2$ incubators (Thermo Fisher Scientific) in VP Serum Free Medium, pH 7.3±0.3 (VP-SFM, Gibco) containing L-glutamine (Gibco).

## Purified EV-A71 antigen and antibodies

To obtain EV-A71 B5 viral particles (E + F particles), Vero cells were infected with HG-B5-141-6-5 at a multiplicity of infection (MOI) = $10^{-4}$. The culture supernatant was collected at 8 days post-infection, followed by filtration through a 0.65 µm filter (Sartorius) and peristaltic pump (Watson-Marlow, filtration speed 100 rpm) to remove cell debris. The filtrate was then inactivated with 0.02% (v/v) formaldehyde for 72 h at 37°C ± 0.3°C. Virus inactivation was verified by the absence of infectious virus in a $TCID_{50}$ (median tissue culture infectious dose) end-point dilution assay. In brief, 1 mL of the inactivated virus culture supernatant was dialyzed against 14 mL of 1 × Dulbecco's phosphate-buffered saline (DPBS) using an Amicon Ultra-15 100K Centrifugal Filter (Millipore) and centrifuged at 3000 rpm, 4°C for 10 min. This cycle was repeated five times to exchange the buffer and obtain the dialyzed supernatant. Next, 1 mL of the dialyzed supernatant was added to pre-seeded Vero cells ($3 × 10^6$ cells in a T25 flask) and incubated for 72 h to monitor the development of the cytopathic effect (CPE). If no CPE was observed, the culture supernatant was harvested and used to inoculate Vero cells for a second passage to monitor for CPE. Supernatant was collected on the 7th day for the $TCID_{50}$ assay to confirm virus inactivation. Following validation of inactivation, the inactivated supernatant was further purified by size-exclusion chromatography using Sepharose 6 Fast Flow resin (GE Healthcare). The experimental parameters were set according to the manufacturer's instructions, with a flow rate of 2 mL/min and 1 × PBS used as the dilution buffer. Finally, the viral antigen fractions were pooled, sterilized using a 0.22 µm filter (Sartorius), and the total protein content was measured using the Pierce BCA Protein Assay Kit (Thermo Fisher Scientific).

MAB979 (Merck) is an antibody against EV-A71 that recognizes the VP0/VP2 capsid protein [20].

## Generation of in-house monoclonal antibody NHRI2016–1

A group of two-month-old, pathogen-free female BALB/c mice (n = 5, body weight = 20.3 g ± 1.22 g) were immunized via intraperitoneal (i.p.) injection with formaldehyde-inactivated B5-141 emulsified in Freund's adjuvant and boosted at two-week intervals. Sera were collected from the 5th to 7th immunizations via the submandibular vein to assess neutralizing antibody responses. Two weeks after the final immunization, splenocytes from immunized mice were fused with the Sp2/0 myeloma cell line, as described previously [22]. The resulting hybridoma supernatants were screened using a neutralization assay against B5-141.

## Generation of escape mutant

B5-141 virus (50 $TCID_{50}$) was diluted and incubated with an equal volume of NHRI2016–1 at 25°C ± 1°C for 1 h. The mixture was then added to an RD cell monolayer in DMEM supplemented with 2% FBS and incubated for 96 h at 37°C ± 0.3°C. If no visible cytopathic effect (CPE) was observed, the culture supernatant was harvested and used to re-inoculate freshly prepared RD cells under the same conditions. This process was repeated until CPE was observed. After six reinfection cycles, the NHRI2016–1 escape mutant was developed and selected, and subsequently named B5-Mut3–1 (GenBank accession number PV176308).

## Western blotting

To examine the EV-A71-specific-mAb-binding sites using western blotting, purified EV-A71 B5 E + F particles were denatured at 98°C for 10 min, separated on a 4–12% SurePAGE Bis-Tris gel (GenScript), and transferred to a polyvinylidene difluoride (PVDF) membrane (Invitrogen) as previously described [18]. Briefly, the membrane was blocked with 5% skim milk diluted in phosphate-buffered saline containing 0.1% Tween 20 (PBST) at 25°C ± 1°C for 1 h. After blocking, the PVDF membrane was incubated with either MAB979 (Merck) or NHRI2016–1 (generated in this study) at a 1:3000 dilution, followed by horseradish peroxidase (HRP)-conjugated anti-mouse IgG secondary antibody (Sigma) at a 1:3000 dilution. BlueRay Prestained Protein Marker (Jena Bioscience) was used as the molecular weight marker.

## RT-PCR, sequencing, and structural analyses

Viral RNA was extracted from the culture supernatant using the RNeasy Plus Mini Kit (QIAGEN). EV-A71 cDNA was synthesized using SuperScript IV Reverse Transcriptase (Invitrogen). Full-length polymerase chain reaction (PCR) amplification and sequencing of EV-A71 were performed using the KAPA HiFi HotStart ReadyMix PCR Kit (Roche Applied Science) and specific primers (S1 Table). The complete genome sequences of the mutant and engineered viruses were verified by Sanger sequencing.

The sequencing results were analyzed using SnapGene 7.0.3. Sequence alignment was performed using BLAST to calculate the amino acid residue identity. A total of fourteen protein sequences (FASTA format) of EV-A71 genogroups and genotypes (GenBank accession numbers: AFN20470.1, BAJ05609.1, AFN20472.1, AFN20473.1, QOP59209.1, QOP59210.1, QOP59198.1, CDN24674.1, AFN20478.1, APW85805.1, AFN20481.1, QIN85535.1, AVQ54963.1, and SJK83382.1) were downloaded from the National Center for Biotechnology Information (NCBI) nucleotide database for sequence comparison. The protein sequences of EV-A71 were aligned using MUSCLE in SnapGene version 7.0.3. The three-dimensional (3D) structure of the human EV-A71 virus (PDB ID: 3VBS) was obtained from the RCSB Protein Data Bank (PDB) database. The locations of specific amino acids on the 3D structure of viral proteins and capsid surface were visualized using PyMOL 2.5. All software were used with default parameters.

## Plasmid constructs

According to the alignment sequencing results between B5-Mut3–1 and B5-141, rgHGB5 was used as the template to generate reverse genetic mutant viruses via site-directed mutagenesis and in-fusion cloning, due to its high titer [18].

VP1 and VP3 viruses were engineered by replacing the VP1 and VP3 regions of rgHGB5 with the corresponding regions of B5-Mut3–1. A single mutation in VP3 of B5-Mut3–1 was introduced into rgHGB5 using site-directed mutagenesis with the KAPA HiFi HotStart ReadyMix PCR Kit (Roche Applied Science), following the manufacturer's instructions (primer sequences listed in S1 Table).

## Virus titration

$TCID_{50}$ was used to determine virus titers in Vero cells. Briefly, a 96-well plate was seeded with $3 \times 10^4$ cells per well in VP-SFM supplemented with L-glutamine and incubated for 24 h at 37°C ± 0.3°C prior to virus infection. Serial dilutions of the virus were prepared in chilled VP-SFM supplemented with L-glutamine. Six replicates of each virus dilution were then added to the 96-well plates and incubated at 37°C ± 0.3°C for 96 h. The 50% endpoint was calculated using the Reed-Muench method [23].

## Immunofluorescence assay (IFA)

The desired $TCID_{50}$ of the virus was added to the Vero cell monolayer in a 24-well plate and incubated at 37°C ± 0.3°C for 48 h. The infected cells were then inactivated with 4% (v/v) formaldehyde at 37°C ± 0.3°C for 30 min, followed by three washes with PBS. Next, 0.1% Triton-X/PBS was added to each well and incubated for 10 min to permeabilize the cells. After removing Triton-X, the fixed cells were blocked with PBS containing 3% bovine serum albumin (BSA) for 1 h. The cells were then washed three times and probed with either MAB979 (Merck) or NHRI2016–1 (1:1000 dilution) for 1 h. An Alexa Fluor 488 goat anti-mouse IgG (H + L) secondary antibody (Invitrogen) (1:200 dilution) was used to detect bound antibodies. The stained samples were examined using an Olympus IX73 inverted microscope (Tokyo, Japan), and images were captured and analyzed using Olympus cellSens software.

## Dot blot assay

Purified EV-A71 B5 E + F particles were processed under heated conditions (90°C, 60°C, 30°C) for 1 h, exposed to UV-C (253.7 nm) for 1 h, or kept unheated (at 4°C). After treatment, the samples were stored at 4°C. A series of two-fold dilutions

of each antigen was loaded onto a nitrocellulose (NC) membrane. After blocking, the NC membrane was probed with either MAB979 (Merck) or NHRI2016–1 (1:3000 dilution), followed by an HRP-conjugated anti-mouse IgG secondary antibody (Sigma) (1:3000 dilution) to detect bound antibodies.

## ELISA

Purified EV-A71 B5 E + F particles were serially diluted in PBS in two-fold dilutions and added to wells pre-coated with a polyclonal antibody derived from VLP (virus-like particle)-immunized rabbit serum (1:4000 dilution). The plate was then incubated at 37°C ± 0.3°C for 2 h. After washing four times, diluted NHRI2016–1 (1:800 dilution) was added to the wells and incubated at 37°C ± 0.3°C for 1 h. The wells were washed four times to remove the primary antibody, and an HRP-conjugated anti-mouse IgG secondary antibody (Sigma) (1:400 dilution) was used as the detecting antibody and incubated at 37°C ± 0.3°C for 1 h. For color development, the plate was incubated with o-phenylenediamine dihydrochloride (OPD) substrate (Sigma) at 25°C ± 1°C for 30 min. After 30 min of incubation, the reaction was terminated with 1 M $H_2SO_4$. Absorbance was measured at $A_{492nm}$.

## Transmission electron microscopy (TEM) analysis

Purified EV-A71 B5 E + F particles were placed onto a carbon film-supported 200-mesh copper grid (Electron Microscopy Sciences) and incubated at 25°C ± 1°C for 2 min. After incubation, excess samples were removed by blotting with filter paper, and the grid was washed twice with deionized water (ddH$_2$O). The E + F particles were then negatively stained by adding 2% phosphotungstic acid (PTA) to the grid for 2 min at 25°C ± 1°C, followed by removal of PTA using filter paper. Finally, the stained grid was dried for 72 h and imaged with a Hitachi H-7650 TEM.

## Rat immunization

NHRI2016–1 was used in the ELISA assay to quantify the ELISA units of purified EV-A71 B5 E + F particles before being processed under heated conditions (90°C, 60°C, 30°C) for 1 h, exposed to UV-C (253.7 nm, G30T8, Sankyo Denki, Japan) for 1 h, or kept unheated (at 4°C) [24]. After treatment, the samples were stored at 4°C for further use.

Seven-week-old, pathogen-free female Wistar rats (body weight = 156 g ± 8.0 g) were used for the immunogenicity assay and were housed in an AAALAC-certified animal facility at the National Health Research Institutes. Groups of rats (n = 4 per group) were vaccinated on days 0 and 14 with EV-A71 B5 E + F particles processed at 90°C, 60°C, 30°C, 4°C (unheated), or exposed to UV-C. Each rat received an intramuscular (i.m.) injection of 400 international units (IU) per dose of the respective antigen, mixed with 30 µg of Adju-Phos adjuvant (InvivoGen) and PBS in a 0.1 mL volume. Blood samples were collected by cardiac puncture on day 28 post-primary immunization. The sera from immunized rats were stored at −20°C for measuring neutralizing antibody titers against HG-B5-141-6-5.

## Neutralization assay

Neutralizing antibody titers were determined as previously described by Chia et al. [21]. In brief, to screen the neutralizing activity of hybridoma supernatants NHRI2016–1 to NHRI2016–5 against B5-141, serial dilutions of each hybridoma supernatant, starting with an initial dilution of 1:8, were two-fold serially diluted, mixed with 100 CCID$_{50}$/50 µL of B5-141 in 96-well plates, and incubated for 1 h at 37°C ± 0.3°C. To measure the neutralizing activities of NHRI2016–1 and MAB979 against different EV-A71 strains (A/BrCr, B4-E59, B5-2008, C4-2008), 50 µL of serially diluted NHRI2016–1 or MAB979 was mixed with 100 CCID$_{50}$/50 µL of A/BrCr, B4-E59, B5-2008, or C4-2008 and incubated at 37°C ± 0.3°C for 1 h in 96-well plates. For the rat study, serum was heat-inactivated at 56°C for 30 min, after which a series of two-fold serial dilutions of the serum were prepared, starting at a dilution of 1:8. The prepared rat serum was mixed with 100 CCID$_{50}$/50 µL of HG-B5-141-6-5 and incubated at 37°C ± 0.3°C for 1 h. The virus-hybridoma supernatant mixture, virus-antibody mixture, or virus-serum mixture was then added to 3 × 10$^4$ RD cells per well and incubated at 37°C ± 0.3°C for 96 h. CPE was

observed using an inverted microscope at 96 h post-infection. Each dilution was performed in triplicate, and the end-point neutralizing titers were defined as the reciprocal of the sample dilutions that inhibited the development of CPE in at least two of three replicates. For each experiment, preserum, positive serum control, and virus back titration were conducted simultaneously. The limit of detection (LD) was set at 1:8.

## Statistical analysis

Data from the animal study, presented as a box-whisker plot, were analyzed using GraphPad Prism 10 and statistical methods. The two-sided Mann-Whitney *U*-test was used to compare the induced antibody titers between 90°C heated and other groups. A P-value less than 0.05 was considered statistically significant.

## Results

### Generation of murine monoclonal antibody NHRI2016–1

Given the urgent need for an EV-A71-specific mAb for in vitro potency assays, we generated hybridoma supernatants by fusing splenocytes from mice immunized with B5-141 with a myeloma cell line (Fig 1A).

The resulting hybridoma supernatants, including NHRI2016–1 to NHRI2016–5 were then tested in a neutralization assay to assess their neutralizing activity. NHRI2016–1 showed a statistically significant difference in neutralizing activity compared to preserum and other hybridoma supernatants (NHRI2016–2 to NHRI2016–5) (Fig 1B). As a result, NHRI2016–1 was selected for further expansion due to its strong reactivity against B5-141.

To rapidly assess the potency of an inactivated EV-A71 vaccine, we aimed to develop a universal mAb capable of recognizing multiple EV-A71 genotypes. A neutralization assay was then conducted to evaluate the cross-reactivity of NHRI2016–1 against several EV-A71 genotypes: A/BrCr, B4, B5, and C4. A/BrCr, a prototype isolated in the 1970s, re-emerged in China in 2008. The other three genotypes are currently predominant EV-A71 strains circulating in the Asia-Pacific region. Notably, NHRI2016–1 could cross-neutralize these tested EV-A71 genotypes (including A/BrCr, B4, B5, C4) with potent neutralization titers, outperforming the commercial antibody MAB979 (Fig 1C).

### Identification of antibody-binding sites

Monoclonal antibody binds to two types of epitopes: linear and conformational forms [25]. Linear epitopes are composed of a single segment of polypeptide chain that commonly refers to a denatured state. In contrast, conformational epitopes are formed by key amino acid residues brought together through protein folding, representing the native state [25].

In western blotting, we found that MAB979 recognized denatured EV-A71 B5 E + F particles, indicating that MAB979 could target the linear form of the antigen (Fig 2A). Meanwhile, NHRI2016–1 did not recognize denatured particles (linear form), suggesting that NHRI2016–1 might recognize the conformational epitopes of the viral capsid (Fig 2B).

Based on the western blotting results, which suggested NHRI2016–1 might target the native state of the antigen, we conducted a co-culture of NHRI2016–1 and EV-A71 virus in RD cells to select escape mutants and identify the critical epitopes recognized by NHRI2016–1. After six rounds of co-culture, most RD cells became infected, indicating that viral replication was no longer inhibited by NHRI2016–1. The viral stock from the NHRI2016–1 escape mutant was collected at the end of the sixth selection cycle, and the mutant was designated B5-Mut3–1 (Fig 2C). We then performed an immunofluorescence assay (IFA) to confirm whether B5-Mut3–1 had escaped recognition by NHRI2016–1. As expected, IFA results showed that there was an apparent green fluorescent signal detected in the B5-141 virus in the presence of either MAB979 or NHRI2016–1, indicating both antibodies could recognize B5-141. In contrast, no green fluorescent signal was detected in B5-Mut3–1 in the presence of NHRI2016–1, suggesting that B5-Mut3–1 had completely abolished NHRI2016–1 recognition (Fig 2D).

To identify the mutations responsible for B5-Mut3–1's escape from NHRI2016–1 recognition, we compared the genomes of B5-141 and B5-Mut3–1. Genomic analysis revealed eight nucleotide mutations in B5-Mut3–1, located in both structural

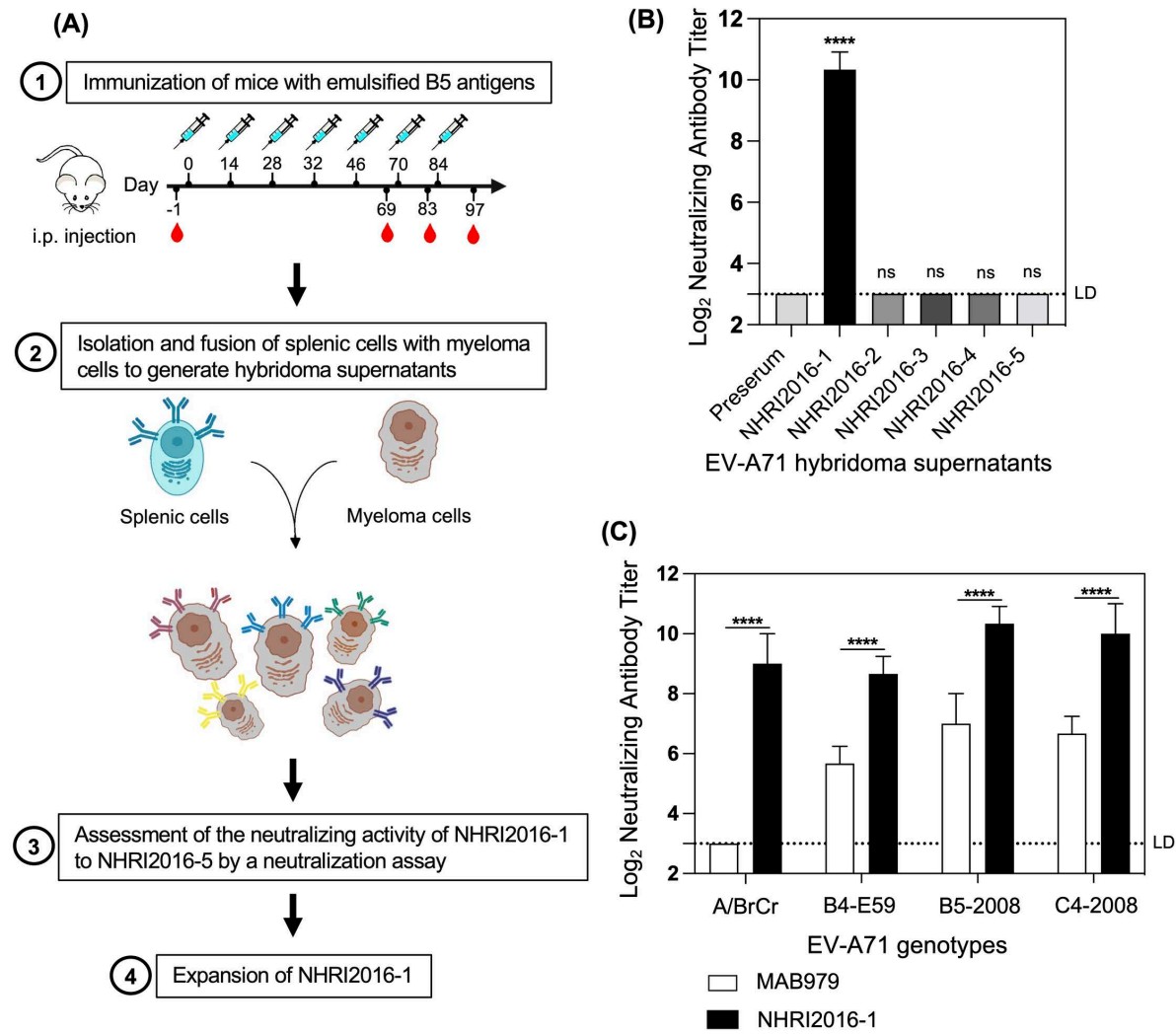

**Fig 1. The neutralizing efficacy of NHRI2016-1** (A) Schematic of EV-A71 hybridoma supernatants generation. After the final boost, splenocytes isolated from immunized mice were fused with a myeloma cell line to create hybridoma supernatants, followed by neutralization assays to select suitable candidates. This schematic is original and created by the authors using Procreate. (B) The neutralizing activity of five hybridoma supernatants (NHRI2016-1 to NHRI2016-5) against B5-141 was assessed by a neutralization assay. Neutralizing titers are presented as the mean ± standard deviation (SD) of triplicates. Statistical significance was determined using a two-way ANOVA, comparing the neutralizing titers between preserum and the hybridoma supernatants (****$P < 0.0001$, ns no significant difference ($P \geq 0.05$)). (C) Neutralizing efficacy of NHRI2016-1 and MAB979 was evaluated against various EV-A71 strains (A/BrCr, B4-E59, B5-2008, C4-2008) by a neutralization assay. Data are presented as the mean ± SD of triplicates. Statistical significance was assessed by two-way ANOVA, comparing the neutralizing titers of MAB979 and NHRI2016-1 against each virus strain (****$P < 0.0001$). LD limit of detection.

and non-structural proteins, compared to B5-141. Since mutations in the non-structural proteins did not contribute to antibody recognition, we focused on the six mutations found in the structural proteins. Among these, one was a synonymous mutation in VP3 (VP3-L101L), and the remaining five were amino acid substitutions in VP1 and VP3, including VP1-N104S, VP1-L183S, VP1-N282D, VP3-S64R, and VP3-Y202C. No mutations were identified in VP2 and VP4 (Table 1).

We further mapped these six mutations onto the EV-A71 pentamer using PyMOL. Remarkably, mutations at VP1-N104S, VP1-N282D, and VP3-S64R were exposed on the EV-A71 capsid surface (Fig 3A). In contrast, VP1-L183S, VP1-L101L, and VP3-Y202C were located on the internal side of the viral capsid (Fig 3B). Therefore, the three mutations

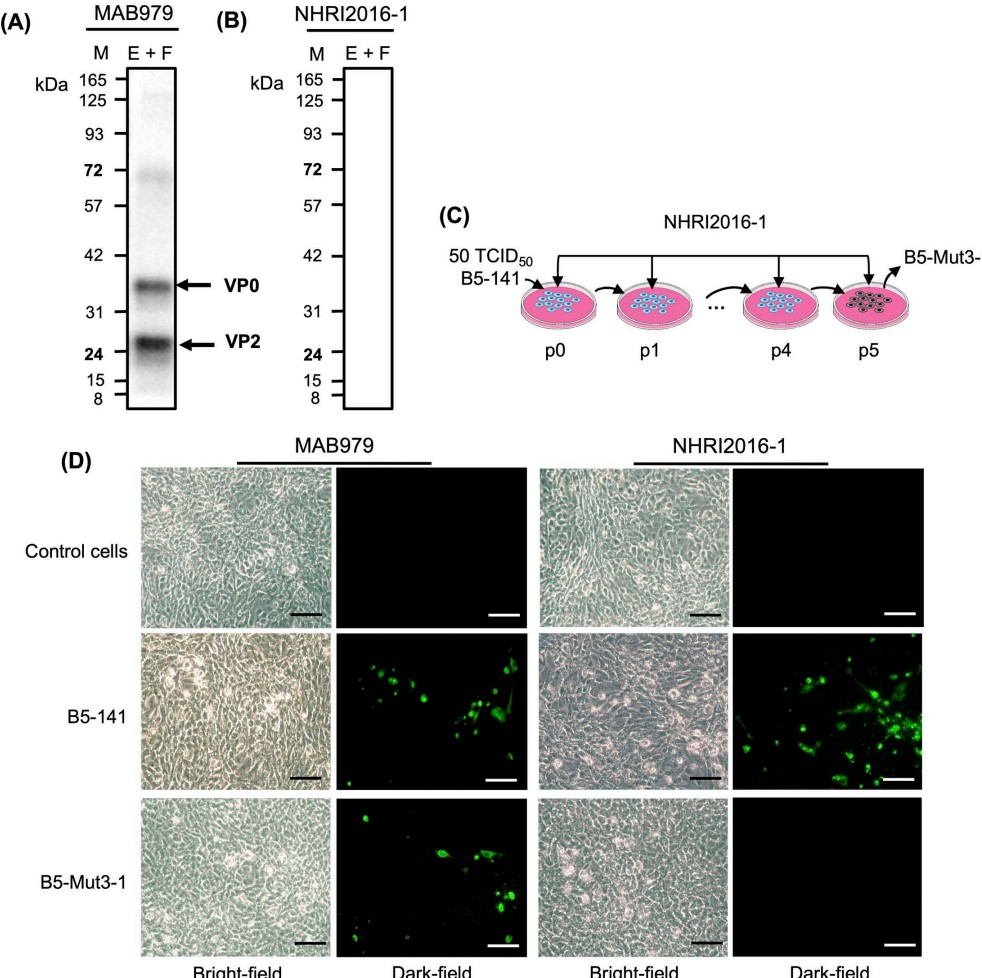

**Fig 2. Characterization of NHRI2016-1 and generation of NHRI2016-1-escape mutant.** (A) Denatured purified EV-A71 B5 E + F particles were detected by western blotting using either MAB979 or (B) NHRI2016-1; MAB979 recognized linear epitopes, while NHRI2016-1 did not. M: marker. (C) Outline of serial passage experiments of B5-141 in the presence of NHRI2016-1. This image is original and created by the authors using Procreate. (D) Binding comparison of MAB979 and NHRI2016-1 to B5-141 and B5-Mut3-1 using IFA. Infected Vero cells were stained with primary antibody (either MAB979 or NHRI2016-1), then incubated with Alexa Fluor 488 goat anti-mouse IgG secondary antibody. NHRI2016-1 did not recognize B5-Mut3-1. Scale bar: 100 μm.

exposed on the EV-A71 capsid surface (VP1-N104S, VP1-N282D, and VP3-S64R) are likely involved in the binding of NHRI2016–1 to B5-Mut3–1.

To identify whether the VP1 or VP3 region of the EV-A71 antigen is responsible for NHRI2016–1 recognition, we used reverse genetic construct rgHGB5 as a template to generate mutant viruses bearing mutations in either VP1 or VP3 of B5-Mut3–1. By substituting the VP1 or VP3 region of B5-Mut3–1 with the corresponding region of rgHGB5, we successfully rescued the VP1 virus, which contained mutations at VP1-N104S, VP1-L183S, and VP1-N282D. However, the VP3 virus, with mutations at VP3-S64R, VP3-L101L, and VP3-Y202C, could not be rescued after three attempts, each involving three passages. We then performed IFA and dot blot to confirm whether NHRI2016–1 could recognize the mutated VP1. A clear signal was detected in both VP1-replaced virus and the positive control rgHGB5 when using

**Table 1. Genetic variations in structural proteins of B5-141 and B5-Mut3-1.**

| Gene | Nucleotide changes | | | Amino acid changes | | |
|---|---|---|---|---|---|---|
| | Position [a] | B5-141 | B5-Mut3–1 | Position [b] | B5-141 | B5-Mut3–1 |
| P1-VP3 | 1908 | T | G | 64 | S | R |
| | 2019 | G | A | 101 | L | L |
| | 2321 | A | G | 202 | Y | C |
| P1-VP1 | 2753 | A | G | 104 | N | S |
| | 2990 | T | C | 183 | L | S |
| | 3286 | A | G | 282 | N | D |

[a] Based on the alignment between B5-141 and B5-Mut3–1 (S1 Fig).

[b] Based on the numbering of the crystal structure of human Enterovirus A71 (PDB ID: 3VBS).

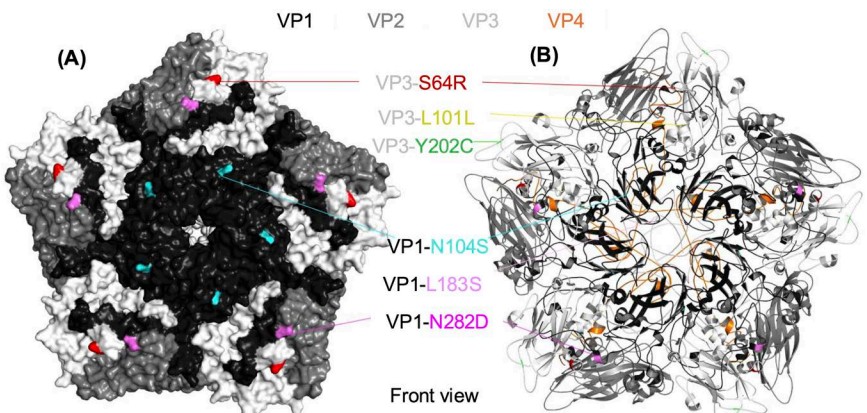

**Fig 3. Epitope mapping of B5-Mut3-1.** The front view of the EV-A71 pentamer is shown using PyMOL. Black, gray, white, and orange represent the EV-A71 viral capsid proteins VP1, VP2, VP3, and VP4, respectively. (A) The surface view showing the location of external mutations. (B) The cartoon diagram of EV-A71 pentamer illustrating both external and internal mutations.

either NHRI2016–1 or MAB979 (Fig 4A and 4B). This result suggested that the VP1 capsid protein was not involved in NHRI2016–1 binding.

Since we could not rescue the VP3-replaced virus, we introduced single mutations from the VP3 region of B5-Mut3–1 into rgHGB5 using site-directed mutagenesis. Single mutations at VP3-S64R, VP3-L101L, and VP3-Y202C were successfully rescued. To examine whether NHRI2016–1 could recognize these viruses, we conducted dot blot and IFA using the same method as previously described. A clear signal was observed in VP3-L101L and VP3-Y202C, but not VP3-S64R, when using NHRI2016–1 (Fig 4A and 4B). Collectively, these data demonstrate that VP3-S64R is associated with NHRI2016–1 binding.

Having observed the critical role of VP3-S64R in NHRI2016–1 recognition in IFA and dot blot, we proceeded to assess whether this position is conserved across other EV-A71 genogroups. The VP3 protein sequences of available EV-A71 genotypes (B1–B5, C1–C5) and genogroups (A/BrCr, D–F) were downloaded from the National Center for Biotechnology Information (NCBI) for comparison. Due to the unavailability of the VP3 sequence for EV-A71 genogroup G in the NCBI database, we could only perform sequence alignment with the available data. The alignment results demonstrated that the binding site of NHRI2016–1 around VP3-S64 region is conserved across all EV-A71 genogroups with available sequences (Fig 5).

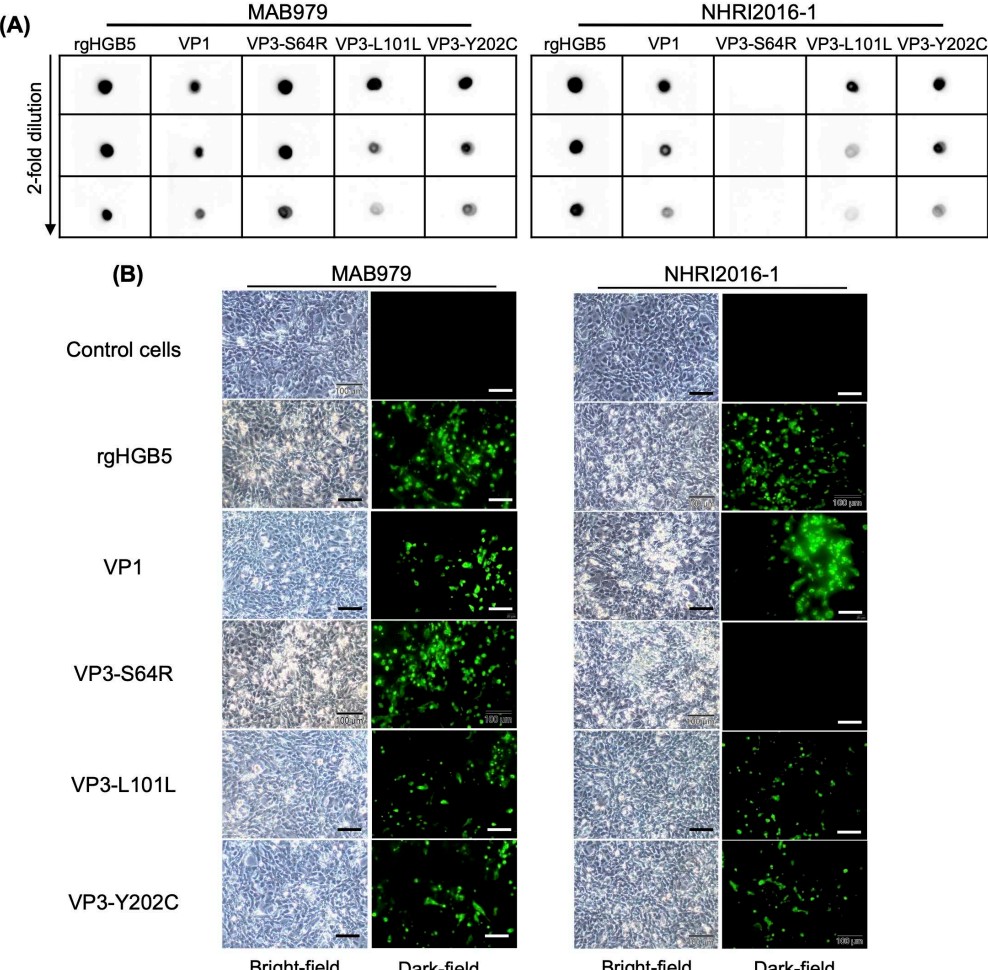

**Fig 4. Mutant virus bearing mutation at VP3-S64R effectively abolishes NHRI2016-1 recognition.** (A) Dot blot data for rgHG-B5 and other mutant viruses. A series of two-fold dilutions of each inactivated virus was loaded onto the nitrocellulose (NC) membrane. After blocking, the NC membrane was probed with either MAB979 or NHRI2016-1, followed by an HRP-conjugated anti-mouse IgG secondary antibody to detect the bound target. (B) Vero cells were infected with either rgHG-B5 or mutant viruses; then, IFA was carried out to detect the green fluorescent signal. Scale bar: 100 μm.

These findings further suggest that NHRI2016–1 could serve as a universal antibody for quantifying and assessing the integrity of most EV-A71 antigens, which is crucial for developing in vitro potency assays for EV-A71 inactivated multivalent vaccines. This is because NHRI2016–1 can recognize a conserved site across different EV-A71 genogroups.

### Determining the correlation between in vitro and in vivo potency assays

Exposure to extreme heat, UV light, and other stressors may lead to a loss of vaccine potency [19]. In this study, NHRI2016–1 was employed in dot blot assay and ELISA to assess the correlation between in vitro and in vivo potency in vaccine samples with varying potency.

To create vaccines with different potency, purified EV-A71 B5 E + F particles were subjected to various conditions: heated at 90°C, 60°C, or 30°C for 1 h, exposed to UV-C (wavelength of 253.7 nm) for 1 h, or kept unheated at 4°C. The samples were then loaded onto NC membranes for the dot blot assay and probed with either MAB979 or NHRI2016–1 (Fig 6A). The dot blot results showed that the antigens exposed to UV-C, heated at 30°C, and kept at 4°C (effective) were

**Fig 5. Sequence comparison of the VP3 region (residues 38 to 77) among multiple EV-A71 genotypes and genogroups.** The VP3-64 residue is highlighted in red, and conserved residues are depicted in black characters on a gray background. The VP3 sequence of the EV-A71 genogroup G is not yet available on NCBI.

recognized by both MAB979 and NHRI2016–1 (Fig 6A). However, NHRI2016–1 failed to detect any signal from the 90°C and 60°C heated (ineffective) antigens, while MAB979 still recognized these samples, though with a slightly decreased signal for the 90°C and 60°C heated (ineffective) antigens. This suggests that the heat treatment may have caused antigen degradation (Fig 6A).

Next, negative-stain transmission electron microscopy (TEM) was performed to examine any morphological changes in the antigens following heat treatment. Interestingly, the 90°C and 60°C heated antigens appeared similar to the 30°C heated, 4°C unheated, and UV-C-exposed antigens, with no evidence of particle disassembly (Fig 6B).

Given that NHRI2016–1 provides distinct measurements for antigens of varying potency in vitro, we assessed the correlation between effective and ineffective antigens and their immunogenicity in animal models. Rats were vaccinated twice with purified EV-A71 antigens–heated at 90°C, 60°C, 30°C, unheated at 4°C, or exposed to UV-C–via intramuscular injection. Serum samples were collected on day 28 after the prime immunization and tested using a neutralization assay to assess antibody titers (Fig 6C). A statistically significant difference in antibody titers was observed between rats vaccinated with effective (30°C heated, 4°C unheated, and UV-C-exposed) vaccines and those vaccinated with ineffective (90°C, 60°C heated) vaccines (Fig 6D). Collectively, these data provide important insights into the functional integrity of the antigens. Specifically, ineffective vaccines lost their integrity and were unable to elicit an immune response, while the effective vaccines successfully induced antibody responses in rats, consistent with the in vitro dot blot assay results. Therefore, NHRI2016–1 can be used to develop in vitro potency assays that quantify effective antigens and assess their stability during vaccine development, manufacturing, and storage.

## Discussion

HFMD is generally mild and self-limiting in most patients. However, some children may develop neurological and systemic complications, or, in rare cases, may even die. Five inactivated monovalent EV-A71 vaccines have been commercially available, with three in China since 2016 and two in Taiwan since 2023. Following the launch of these vaccines, the proportion of severe and fatal HFMD cases in China has sharply decreased by 62.20% and 83.78%, respectively [15]. This decline indicates that the monovalent EV-A71 vaccines have had a significant positive impact on controlling HFMD [15]. As these vaccines are implemented, it is necessary to continuously assess the quality and integrity of each vaccine batch.

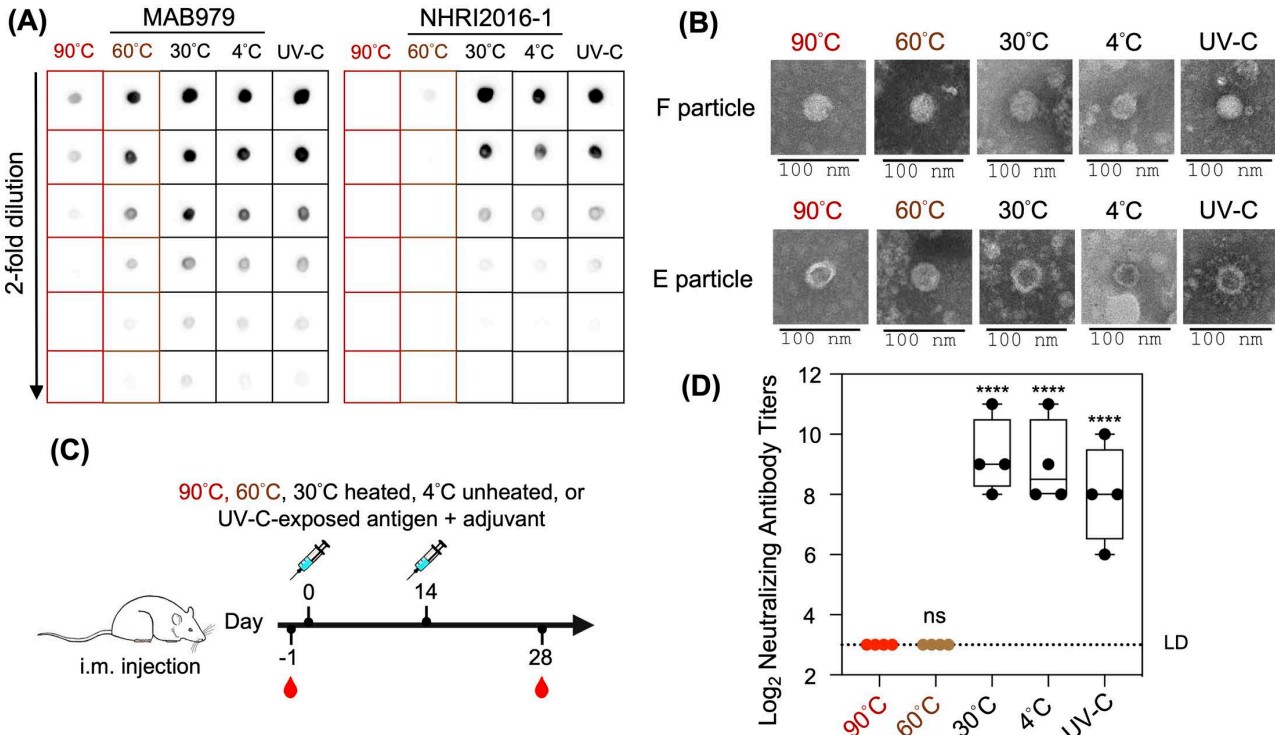

**Fig 6. Dot blot data and rat immunization study using NHRI2016-1 to quantify different EV-A71 immunogenic forms.** Ineffective antigens refer to those that have lost their integrity (e.g., 90°C and 60°C heated antigens). Effective antigens maintain their potency and integrity (e.g., 30°C heated, 4°C unheated, UV-C exposed). (A) Dot blot results of antigens recognized by MAB979 compared to NHRI2016-1. Red boxes: 90°C heated antigen; brown boxes: 60°C heated antigen; black boxes: 30°C heated, 4°C unheated and UV-C-exposed antigens. (B) TEM images of effective and ineffective antigens. (C) NHRI2016-1 was employed in ELISA to quantify antigen concentrations. Specific pathogen-free female Wistar rats (n = 4 for each group) were immunized with a prime dose on day 0 and a booster injection on day 14, with 400 IU of the respective antigens. This schematic is original and created by the authors using Procreate. (D) Immunogenicity of effective and ineffective antigens in rats at day 28 post-primary immunization against homologous virus. LD: limit of detection. The box and whisker diagram represents neutralizing antibody titers for each group. The bar in the box indicates median antibody titers, and the whiskers range from the minimum to the maximum titer. Each dot represents the antibody titer of each rat. A two-sided Mann-Whiney U-test showed a significant difference in antibody titer between rats immunized with ineffective (90°C and 60°C heated) and effective (30°C heated, 4°C unheated, and UV-C-exposed) antigens (****P < 0.0001, ns no significant difference (P ≥ 0.05)).

In this context, antibodies play a key role in evaluating the potency of both licensed and under-development vaccines. Traditionally, in the absence of such antibodies, potency is assessed using in vivo animal models (e.g., mice, rats), where animals are challenged with the vaccine antigen and their immune responses are analyzed. While these in vivo assays serve as surrogates for vaccine efficacy, they can be time-consuming, costly, and raise ethical concerns. In contrasts, in vitro assays are more efficient, require fewer reagents, and can be performed with laboratory personnel, thus reducing operational costs. However, to date, no commercially available antibody specific to EV-A71 has been developed for use in in vitro potency assays.

Like other members of the *Enterovirus* genus, such as human rhinoviruses and polioviruses, EV-A71 undergoes conformational changes, converting from native (F particles) to expanded virions (E particles) upon the delivery of its viral genome to the host cytosol [26,27]. These conformational changes can also be triggered by acidic pH [27]. A previous study suggested that both virus particles are immunogenic [18]. Moreover, exposing vaccines to environmental conditions such as extreme heat, freeze-thaw cycles, or UV light can increase the risk of compromising vaccine potency and effectiveness [28]. Therefore, developing effective vaccine potency tests is crucial, not only to minimize the financial risk

associated with vaccine wastage due to compromised quality but also to ensure the success of national immunization programs. Potency serves as an indicator of vaccine stability, reflecting how environmental factors might impact immunocompetence. Immunocompetence refers to the immunogenicity of a vaccine and its ability to stimulate subsequent protective efficacy [28,29]. Potency tests are integral throughout vaccine development, manufacturing, and storage [28,29]. Stress testing under extreme environmental conditions, such as heat or UV light exposure, helps determine whether potency assays accurately reflect the functionality of the vaccine [28,29].

In this study, we processed the EV-A71 vaccine, which consists of both native (F particles) and expanded (E particles) virions, under various conditions: heated at 90°C, 60°C, or 30°C for 1h, exposed to UV-C for 1h, or kept unheated at 4°C. We then employed our in-house monoclonal antibody, NHRI2016–1, and the commercial antibody MAB979 in a dot blot assay to compare their ability to distinguish between effective (heated at 30°C, exposed to UV-C for 1h, or kept unheated at 4°C) and ineffective (heated at 90°C or 60°C for 1h) antigens. The dot blot data showed that NHRI2016–1 exclusively recognized effective antigens, while MAB979 could recognize all antigens of varying potency (Fig 6A). Since MAB979 targets a linear epitope formed by VP2 residues 141–150, it is possible that this epitope remains exposed and intact, albeit slightly altered, under extreme heat treatment at 90°C and 60°C [20]. As a result, MAB979 is still able to bind to these ineffective antigens, though with reduced efficacy compared to the effective ones.

When effective and ineffective antigens were tested in an in vivo potency test using a rat model, only the effective antigens elicited immune responses, while the ineffective antigens did not (Fig 6C and 6D). Since NHRI2016–1 used in in vitro assays (e.g., ELISA, dot blot) could accurately predict in vivo vaccine effectiveness in a rat model (Fig 6A and 6D), these data offer several advantages, aligning with the 3Rs principles (Replacement, Reduction, and Refinement). This in vitro and in vivo correlation could reduce the need for animal models and minimize the number of animals used by screening degraded, ineffective vaccines in vitro before proceeding to preclinical trials.

In addition, the UV-C spectrum (200–280 nm) is widely recognized as the germicidal range for UV irradiation. Within this range, viral nucleic acids (including RNA and DNA) absorb UV-C photons, causing significant damage. The absorbed photons induce crosslinking between pyrimidine bases in viral RNA, forming dimers that hinder proper replication or translation. Therefore, the effect of UV irradiation is referred to as "inactivation" rather than "killing" [30]. Given that UV-C may also affect vaccine potency, we exposed the antigen to UV-C for 1h and tested it both in vitro and in vivo (Fig 6A and 6D). We found that the UV-C-exposed antigen was still recognized in vitro by both MAB979 and NHRI2016–1 (Fig 6A) and was able to induce an immune response in rats (Fig 6D), similar to the responses observed for antigens heated at 30°C or kept unheated at 4°C. This suggests that UV-C irradiation did not destroy the conformation of the inactivated whole virion vaccine, as observed in TEM analysis (Fig 6B). Previous studies have also suggested that UV irradiation insignificantly contributes to capsid damage in both enveloped and non-enveloped viruses [31,32]. Since UV irradiation targets viral nucleic acids, it may have a detrimental effect on DNA, RNA or live attenuated vaccines, but not on inactivated whole virion vaccines.

Several structural studies have described how EV-A71 attaches to the cell and undergoes endocytosis. Neutralization process can occur at multiple steps, including restricting viral entry, uncoating, and genome release [17,33]. Neutralizing antibodies typically target the structural epitopes located in the viral capsid, some of which may overlap with cellular receptor-binding sites of EV-A71. EV-A71 utilizes multiple functional receptors for cell entry, such as PSGL1 (P-selectin glycoprotein ligand 1), annexin II, heparan sulfate, sialic acid, and SCARB2 (scavenger receptor class B member 2) for viral uncoating [17]. The SCARB2 receptor recognizes the southern rim of the canyon, where it engages the VP1-GH loop and the VP2-EF loop [17,34]. The murine monoclonal antibody 22A12 binds to VP1 epitopes in the southern rim of the canyon, which overlaps with SCARB2 binding sites [35,36]. The binding sites of for PSGL1 and heparan sulfate overlap significantly, with both binding at the 5-fold vertex [34]. PSGL1 recognizes VP1 residues 145, 242, and 244, while heparan sulfate binds to VP1 residues 162, 242, and 244. In contrast, the binding sites for annexin II and sialic acid remain unknown [36]. In this study, we identified the neutralization epitope of NHRI2016–1, located in the VP3 "knob" region

around residue 64. Similarly, mAb 10D3 could bind to VP3 residues 59, 62, and 67 [37], and Plevka et al. demonstrated that mAb E19 also primarily recognizes the VP3 "knob" [33]. The neutralizing epitopes of NHRI2016–1, 10D3, and E19 are not located within or adjacent to any known receptor-binding sites [33,34,37]. It is possible that NHRI2016–1 neutralizes EV-A71 by preventing its binding to as-yet-unidentified receptors.

Although we identified VP3-S64R as a key amino acid involved in the binding interaction between NHRI2016–1 and EV-A71, further structural studies (e.g., Cryo-EM) are needed to fully define the complete binding site and the precise conformational arrangement of the amino acids involved. Another consideration is that we exposed the antigen to extreme temperatures (90°C and 60°C for 1 h) to generate ineffective vaccines. It is important to note that vaccines are not typically exposed to such high temperatures during normal storage and transport. The heat exposure in our study served as a stress test for the vaccine's integrity, providing a clear benchmark for its resilience to temperature-induced degradation. This extreme condition helps assess the robustness of vaccines and the potential impact of accidental exposure to high temperatures during transit or improper storage. Our primary aim was to test NHRI2016–1's ability to differentiate vaccine quality in a relatively short time frame, and the extreme heat exposure allowed us to rapidly and effectively assess this capability. Given that NHRI2016–1 successfully distinguishes between effective (30°C heated, 4°C unheated, and UV-C-exposed) and ineffective (90°C, 60°C heated) antigens, further studies should evaluate antigens stored at room temperature (e.g., 25°C or 30°C) or at 4°C for extended periods (months or years). Previous studies have shown that the formalin-inactivated EV-A71 vaccine maintain stability and potency for over 18 months at 4°C [38]. Meanwhile, inactivated poliovirus vaccines (IPV) also demonstrate stable potency at 4°C for up to 3 years. However, these IPVs lose their potency quickly if exposed to temperatures above room temperature (25°C) or undergo freezing [39,40]. Long-term stability testing under more realistic condition, such as those encountered during vaccine manufacturing, storage, and shipping (especially in areas with limited cold-chain facilities), would provide valuable insights into the stability of EV-A71 vaccines.

In summary, our work demonstrates that the NHRI2016–1 antibody exclusively recognizes effective antigens in vitro, which directly correlates with in vivo potency in a rat model. NHRI2016–1 targets the conformational epitope around S64 in the VP3 "knob" region, which is highly conserved across all available EV-A71 genogroups. Therefore, NHRI2016–1 could serve as a potential tool for monitoring the potency of both monovalent and multivalent EV-A71 vaccines. These findings may have significant implications for the development of EV-A71 vaccines.

## Supporting information

**S1 Table. Primers used in PCR and sequencing.**
(XLSX)

**S2 Table. Raw data for building the graphs in this study.**
(XLSX)

**S1 Fig. The whole genome sequence alignment between B5-141 and B5-Mut3–1 was performed using Local alignment (Smith-Waterman).** Consensus nucleotides are depicted in black, while non-consensus nucleotides are highlighted in red.
(TIFF)

## Acknowledgments

We thank Mark Swofford for manuscript editing. THLL is currently pursuing a doctoral degree in the Graduate Program of Biotechnology in Medicine, a joint program offered by National Tsing Hua University and the National Health Research Institutes.

## Author contributions

**Conceptualization:** Thi-Hong-Loc Le, Tzu-Yu Weng, Hua Yen, Min-Yuan Chia, Min-Shi Lee.

**Data curation:** Thi-Hong-Loc Le.

**Formal analysis:** Thi-Hong-Loc Le.

**Funding acquisition:** Min-Shi Lee.

**Investigation:** Thi-Hong-Loc Le.

**Methodology:** Tzu-Yu Weng, Hua Yen, Min-Yuan Chia.

**Supervision:** Tzu-Yu Weng, Min-Shi Lee.

**Visualization:** Thi-Hong-Loc Le, Tzu-Yu Weng.

**Writing – original draft:** Thi-Hong-Loc Le.

**Writing – review & editing:** Thi-Hong-Loc Le, Tzu-Yu Weng, Min-Yuan Chia, Min-Shi Lee.

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
