## [Decision Letter · Decision Letter 0]

Development of a novel EV-A71 monoclonal antibody for monitoring vaccine potency

Dear Dr. Lee,

Thank you for submitting your manuscript to PLOS Neglected Tropical Diseases. After careful consideration, we feel that it has merit but does not fully meet PLOS Neglected Tropical Diseases's publication criteria as it currently stands. Therefore, we invite you to submit a revised version of the manuscript that addresses the points raised during the review process.

Please submit your revised manuscript within 60 days Mar 29 2025 11:59PM. If you will need more time than this to complete your revisions, please reply to this message or contact the journal office at plosntds@plos.org. Please include the following items when submitting your revised manuscript:

We look forward to receiving your revised manuscript.

Kind regards,

Ran Wang, M.D.

Academic Editor

Shaden Kamhawi

Editor-in-Chief

Shaden Kamhawi

co-Editor-in-Chief

Paul Brindley

co-Editor-in-Chief

**Additional Editor Comments (if provided):**

**Journal Requirements:**

**Reviewers' Comments:**

Reviewer's Responses to Questions

**Key Review Criteria Required for Acceptance?**

**Methods**

-Are the objectives of the study clearly articulated with a clear testable hypothesis stated?

-Is the study design appropriate to address the stated objectives?

-Is the population clearly described and appropriate for the hypothesis being tested?

-Is the sample size sufficient to ensure adequate power to address the hypothesis being tested?

-Were correct statistical analysis used to support conclusions?

-Are there concerns about ethical or regulatory requirements being met?

Reviewer #1: Yes, to all above except last bullet point. No ethical or regulatory concerns.

Reviewer #2: The methods are complete and the technical system is well-established.

Reviewer #3: (No Response)

Reviewer #4: (No Response)

**Results**

-Does the analysis presented match the analysis plan?

-Are the results clearly and completely presented?

-Are the figures (Tables, Images) of sufficient quality for clarity?

Reviewer #1: Yes

Reviewer #2: 1. Figure 1 could include images of mouse inoculation to make the article more vivid and intuitive.

2. In the potency assays section, results under UV exposure conditions should also be added.

Reviewer #3: The authors developed a monoclonal antibody (NHRI2016-1) that might be used to evaluate the potency of EV-A71 vaccines either in vitro or in vivo. They clearly showed processes of the generation, the binding site identification and the evaluation of NHRI2016-1. This is a great work. However, there are some issues need to be addressed.

Major comments:

1. How was the B5-141 that was employed to generate monoclonal antibodies inactivated? If it was inactivated by heating, it would be identified by NHRI2016-1 after heating in Result Ⅲ.

2. Please provide additional experimental data on the in vivo effectiveness of the vaccine at various temperatures (4℃, 30℃, 60℃ and 90℃).

3. Vaccines are rarely kept at 90℃ in the real world. However, NHRI2016-1 can only distinguish between effective and ineffective vaccines at 90℃. Please give the practical significance of this result.

Minor comments:

1. Line 44. Please give the full name of HFMD.

2. Line 80. It is recommended to add the reason that MAB979 failing to distinguish between effective and ineffective vaccines.

3. Line 163. Please show the sequence or reference of specific primers.

4. Line 277. Why don’t the authors evaluate the genotypes D, E, F and G?

5. Table 1. It is recommended to use a bar chart to display the results of this part, which is more intuitive

6. Line 271. Please add the results of screening to Supplementary.

7. Line 303. It is recommended to add the sequence of B5-Mut3-1 to GenBank.

8. Can UV render ineffective vaccines undetectable by NHRI2016-1?

Reviewer #4: (No Response)

**Conclusions**

-Are the conclusions supported by the data presented?

-Are the limitations of analysis clearly described?

-Do the authors discuss how these data can be helpful to advance our understanding of the topic under study?

-Is public health relevance addressed?

Reviewer #1: Yes

Reviewer #2: The experimental results support the conclusion.

Reviewer #3: (No Response)

Reviewer #4: (No Response)

**Editorial and Data Presentation Modifications?**

Reviewer #1: Refer to general comments section below

Reviewer #2: (No Response)

Reviewer #3: (No Response)

Reviewer #4: (No Response)

**Summary and General Comments**

Reviewer #1: 1. Abstract: It might be worth including that EV-A71 causes FMHD in the abstract. Its included in the next section (author summary), but I recommend it including it in the abstract for the benefit of the reader.

2. Introduction: Lines 89-91: “We also identified the binding epitope of NHRI2016-1, which in turn may have important implications for EV-A71 vaccine development”. Based on the experiments and the results, it seems that one conserved amino acid on the antigen (VP3-S64) was identified to be key in binding of the mAb to the antigen. Is it fair to claim that you have identified the “epitope”, when multiple amino acids located in different regions come together to form the conformational epitope that would potentially be involved in the binding of the mAb to the antigen? You might have to employ CryoEM to get a complete picture of the multiple amino acids involved to form the epitope. Please clarify.

3. Neutralization Assay (Methods section) and Table 1 data: How was the cross reactivity evaluated. In the methods section, the authors mention that HG-B5-141-6-5 which is the B5 genotype. What about the others?

4. Results section, Lines 269-273: “Given the urgent need for EV-A71-specific mAb used in in vitro potency assays, we generated hybridoma clones using isolated splenocytes from mice immunized with inactivated EV-A71 genotype B5 (B5-141). After screening by dot-blot and neutralization assays, mAb NHRI2016-1 was selected for further expansion among five mAbs due to its strong reaction against B5-141”. A bit more background and limited key data/results regarding how this final mAb was selected would be beneficial to the readers.

5. Results section, lines 303-306: “To test whether B5-Mut3-1 indeed escaped from NHRI2016-1 recognition, we freeze-thawed three times prior to determining the virus titer and performing an immunofluorescence assay (IFA)”. Can you clarify what was Freeze/Thawed? Virus or the mAb?

6. Fig 3d IFA data: are the viral particles expected to be native (nondenatured) during the IFA assay. If yes, then why would MAB979 which binds to a linear epitope show a green signal with either the B5-141 or B5-Mut3-1. Is it hypothesized that the linear epitope to which MAB979 binds is on the surface of the viral particle?

7. Table 2: raw data for the genome sequence analysis are not provided.

8. Data in Figure 3A: Does the intensity of the signal (compared to the control rgHGB5) tell us anything about the mAb binding affinity or is it just all or none phenomena as in the case of NHRI2016-1 and VP3-S64R mutant. For example, compared to the rgHGB5 control, the binding of NHRI2016-1 to the VP1, VP3-L101L, VP3Y202C is markedly different at most dilutions tested. Does lower binding in dot blot for a particular mutation indicate the role of that amino acid in the binding of the mAb to the antigen. Please comment.

9. Results section, Line 402-405: “First, to create a vaccine with different potency, purified EV-A71 B5 E + F particles were processed under heated (at 90 °C, 60 °C, 30 °C) or unheated (kept at 4 °C) conditions, loaded onto NC membrane in dot-blot assay, and probed with either MAB979 or NHRI2016-1 (Fig 5A)”. The duration used for the high temperature exposure is not specified.

10. Related to my earlier comment regarding the location of the linear epitope to which mAb979 binds (surface exposed or buried), how do you explain the loss in binding seen for mAb979 after exposure to 90C (signal is lower compared to 4C). If the epitope is linear and exposed (which seems likely because binding is observed at 4C), the binding should potentially not change.

11. Figure 5b TEM data: Was aggregation (clumping) of the antigen observed in the 90C sample?

12. Please comment on why 90C incubation was deemed relevant to show in vitro in vivo correlation. Vaccines will never get exposed to such a high temperature. Instead, why not choose an extended incubation at 25 or 30C for several months (3 or 6 months) to demonstrate the correlation under more relevant conditions that the vaccine may be exposed to during manufacturing, storage, and especially during shipment to areas where there is a lack of cold chain facilities (rural regions).

13. For the in vivo experiments, it’s definitely possible that there was confirmational change in the antigen after 90C exposure and there a response was not seen after injecting in mice. However, there is a possibility that the antigen was complete aggregated and might have fallen out of solution after 90C exposure (formation of visible particles), which would change the concentration of the antigen in solution. How did you ensure that there was no aggregation, and you injected the same amount of antigen (400 IU) into the animal for both the 90C and the 4C arm?

14. Figure 5d: If I understand correctly, the control shown in the whisker plot was an antibody used in the neutralization assay and was not an additional arm in the in vivo experiment? What sample was used in the neutralization assay with this antibody? Please provide additional context regarding the control in the main text.

15. Any reason why the neutralization titers of NHRI2016-1 were not compared to that of MAB797 in the in vivo experiments (Fig 5D).

16. Context regarding how the potency is currently measured in the absence of an antibody like NHRI2016-1 is missing. What is the release potency assay? If potency is currently measured using an in vivo assay due to the lack of an antibody like NHRI2016-1, provide advantages of using the mAb in a dot blot assay or ELISA in terms of time and cost saving. Additionally, consider discussing how the mAb will aid in the Replacement, reduction, and refinement efforts in vaccine potency measurements.

Reviewer #2: The experimental and result sections of this article are relatively complete and well-developed, but the introduction requires improvement and further consideration. In China, EV-A71 has not been detected for a long time, while CV-A6/16/10 has become the predominant circulating strain and is associated with some cases of pediatric encephalitis. Therefore, the significance of EV-A71 presented in the introduction is relatively weak. It is recommended that the authors provide more information on the vaccination and application status of the EV-A71 vaccine.

Reviewer #3: (No Response)

Reviewer #4: (No Response)

**Figure resubmission:**

**Reproducibility:**



---

## [Decision Letter · Decision Letter 1]

Dear Dr. Lee,

We are pleased to inform you that your manuscript 'Development of a novel EV-A71 monoclonal antibody for monitoring vaccine potency' has been provisionally accepted for publication in PLOS Neglected Tropical Diseases.

Best regards,

Ran Wang, M.D.

Academic Editor

Shaden Kamhawi

Editor-in-Chief

Shaden Kamhawi

co-Editor-in-Chief

Paul Brindley

co-Editor-in-Chief

Reviewer's Responses to Questions

**Key Review Criteria Required for Acceptance?**

**Methods**

-Are the objectives of the study clearly articulated with a clear testable hypothesis stated?

-Is the study design appropriate to address the stated objectives?

-Is the population clearly described and appropriate for the hypothesis being tested?

-Is the sample size sufficient to ensure adequate power to address the hypothesis being tested?

-Were correct statistical analysis used to support conclusions?

-Are there concerns about ethical or regulatory requirements being met?

Reviewer #2: (No Response)

Reviewer #3: (No Response)

Reviewer #4: (No Response)

**Results**

-Does the analysis presented match the analysis plan?

-Are the results clearly and completely presented?

-Are the figures (Tables, Images) of sufficient quality for clarity?

Reviewer #2: (No Response)

Reviewer #3: (No Response)

Reviewer #4: (No Response)

**Conclusions**

-Are the conclusions supported by the data presented?

-Are the limitations of analysis clearly described?

-Do the authors discuss how these data can be helpful to advance our understanding of the topic under study?

-Is public health relevance addressed?

Reviewer #2: (No Response)

Reviewer #3: (No Response)

Reviewer #4: (No Response)

**Editorial and Data Presentation Modifications?**

Reviewer #2: (No Response)

Reviewer #3: (No Response)

Reviewer #4: (No Response)

**Summary and General Comments**

Reviewer #2: (No Response)

Reviewer #3: (No Response)

Reviewer #4: (No Response)

PLOS authors have the option to publish the peer review history of their article (what does this mean? ). If published, this will include your full peer review and any attached files.

**Do you want your identity to be public for this peer review?** For information about this choice, including consent withdrawal, please see our Privacy Policy .

Reviewer #2: No

Reviewer #3: No

Reviewer #4: No

---

## [Editor Report · Acceptance letter]

Dear Dr. Lee,

We are delighted to inform you that your manuscript, "Development of a novel EV-A71 monoclonal antibody for monitoring vaccine potency," has been formally accepted for publication in PLOS Neglected Tropical Diseases.

Best regards,

Shaden Kamhawi

co-Editor-in-Chief

Paul Brindley

co-Editor-in-Chief
